# Neuroinflammation: A Signature or a Cause of Epilepsy?

**DOI:** 10.3390/ijms22136981

**Published:** 2021-06-29

**Authors:** Enrico Pracucci, Vinoshene Pillai, Didi Lamers, Riccardo Parra, Silvia Landi

**Affiliations:** 1National Enterprise for Nanoscience and Nanotechnology (NEST), Istituto Nanoscienze Consiglio Nazionale delle Ricerche (CNR) and Scuola Normale Superiore Pisa, 56127 Pisa, Italy; enrico.pracucci@sns.it (E.P.); vinoshene.pillai@sns.it (V.P.); didi.lamers@ru.nl (D.L.); riccardo.parra@gmail.com (R.P.); 2Institute of Neuroscience CNR, 56127 Pisa, Italy

**Keywords:** epilepsy, neuroinflammation, brain excitability

## Abstract

Epilepsy can be both a primary pathology and a secondary effect of many neurological conditions. Many papers show that neuroinflammation is a product of epilepsy, and that in pathological conditions characterized by neuroinflammation, there is a higher probability to develop epilepsy. However, the bidirectional mechanism of the reciprocal interaction between epilepsy and neuroinflammation remains to be fully understood. Here, we attempt to explore and discuss the relationship between epilepsy and inflammation in some paradigmatic neurological and systemic disorders associated with epilepsy. In particular, we have chosen one representative form of epilepsy for each one of its actual known etiologies. A better understanding of the mechanistic link between neuroinflammation and epilepsy would be important to improve subject-based therapies, both for prophylaxis and for the treatment of epilepsy.

## 1. Introduction

Neuroinflammation is the process of inflammation that involves nervous tissues, and it can be originated by several exogenous or endogenous factors [1,2,3]. Several factors can activate neuroinflammation, such as infection, traumatic brain injury, toxic metabolites, autoimmune diseases, aging, air pollution, passive smoke or spinal cord injury, and stimulate the production of cytokines and chemokines, which also act as a support for cell growth and survival. They include at least 40 types of interleukins (IL), first thought to be expressed only by leukocytes, but later found to be produced by different cell types [4]. Cytokines and chemokines activate microglia, as a primary immune response in the central nervous system (CNS). Continuous microglia activation causes the recruitment of peripheral immune cells [5], such as macrophages and B and T lymphocytes, which are responsible for the innate and adaptive immune response. These immune cells can access the brain through a compromised blood brain barrier (BBB), amplifying the defense mechanism and bringing about widespread chronic inflammation, and possibly neurodegenerative effects [6]. Another cellular component activated during neuroinflammation is represented by astrocytes; they are strictly linked to the BBB structure and can be responsive to signals released by injured neurons or activated microglia. Their contribution to tissue repair can be substantial, as in the case of glial scar formation, which is retained to promote axonal regeneration [1]. However, prolonged chronic insults can favor the activation of molecular pathways that sustain the inflammatory properties from brain-resident cells, causing a maladaptive response that can be harmful to the CNS [7].

Many studies have explored the interaction between neuroinflammation and neurological disorders, particularly with epilepsy [7,8]. Epilepsy can be a primary pathology, due to structural or genetic reasons, or a secondary effect. In the latter case, it can be a consequence of traumatic brain injuries and brain tumors; then, it can be related to an infectious, metabolic, immune or unknown etiology, as summarized in the last ILAE classification of the epilepsies [9]. Undoubtedly, the presence of certain chronic inflammatory diseases facilitates epilepsy or other neurological manifestations. Indeed, in most autoimmune diseases, there is a five-fold increased risk of epilepsy in children and a four-fold increased risk in non-elderly adults (aged < 65) [10,11]. Even though the impaired regulation of the inflammatory response in injured neuronal tissue is a critical factor to the development of epilepsy, it is still unclear how **this** unbalanced regulation of inflammation contributes to epilepsy [8]. On the other hand, several studies have shown that epileptogenesis produces long-term effects on neuroinflammation, worsening the progression and outcome of epilepsy [7,8,12,13,14].

In these last years, common pathways relating epilepsy to neuroinflammation have been identified, starting from the pioneering study of Goddard [14,15,16,17]. Interestingly, different models of chemically and electrically induced seizures show upregulation of genes expressed in inflammatory cascades, as seen in patients [18]. In epileptic rodent models, a key role is played by IL-1β, its receptor (IL-1R), and the antagonist of its receptor (IL-1Ra) [18-19-20-21-22-44]. Epileptogenesis, as well as several other conditions that bring about secondary epileptic phenotype [19,20,21], is also correlated to the activation of Toll-like receptors (TLRs). Indeed, TLRs are responsible for the innate immune response, as factors upstream of IL-1β. Once a pathogen enters the organism, transmembrane receptors that are especially present on the membrane of macrophage and dendritic cells, recognizes it, and triggers localized inflammation. Moreover, various hyperacetylated molecules, such as “high-mobility box 1 group protein” (HMGB), a chromatin component released during necrosis, are capable to amplify TLRs activation [22], and are involved in ictogenesis in models of chronic epilepsy and in humans [23]. Further factors are tumor necrosis factor- alfa (TNF-α), transforming growth factor beta (TGF-β), cyclo-oxygenase 2, and thrombospondin (TSP-1) [24]. Recently, the pentraxin family (PTXs) has also been identified to be involved in the immune response promoting epilepsy. PTX3 is expressed in the brain, where it is secreted by several white blood cells in response to inflammatory signals [25]. It interacts with the extracellular matrix and participates in remodeling AMPA receptors, regulating circuit excitability. PTX3 activation has been shown to have a pivotal role in a mouse model of experimental autoimmune encephalitis [26]. Then, it has been shown that the upregulation of inflammation causes effects at the extracellular matrix level, increasing levels of the redox-sensitive matrix metalloproteinase MMP-9 inside the epileptic brain [27] and in schizophrenia [28]. MMP9 stimulates the receptor for advanced glycation end-products (RAGE), eventually leading to the secretion of various cytokines; changes in the extracellular matrix can finally impinge on the balance between excitation and inhibition, and on synaptic plasticity [29,30]. Another novel mechanism reinforcing neuroinflammation is supported by the renin-angiotensin system (RAS), which reinforces immune system activation; blocking this pathway prevents neurobehavioral effects of neuroinflammation, induced by lipopolysaccharide (LPS) treatment [31].

Considering this background, the goal of our review is to show some of the known aspects of the mechanistic relationship between neuroinflammation and epilepsy, mainly focusing on certain paradigmatic diseases as focal cortical dysplasia, PCDH19 epilepsy, glioblastoma multiforme (GBM), maternal immune activation, multiple sclerosis, autism spectrum disorders (ASD) associated with epilepsy, and SARS-COV-2 (Figure 1). Following this, we finally explore several therapies that are currently being employed in epileptic patients, targeting neuroinflammation.

## 2. Mechanisms of Neuroinflammation in Some Exemplificative Pathologies Related to Different Forms of Epilepsy

### 2.1. Structural Epilepsy: Focal Cortical Dysplasia

Focal cortical dysplasia (FCD) is due to an anomaly in cortical development, and it is among the first causes of drug-resistant epilepsy in children and in adults [32]. Several typologies of FCD have been classified according to their peculiar anatomical and functional alterations [33]. These alterations, characterized by the emergence of balloon cells or disrupted lamination, or by the presence of ectopic neurons, are often caused by genetic mosaic mutations of genes involved in **the** mTOR pathway [33,34].

Such neuronal abnormalities are accompanied by neuroinflammation, and the degree of activated microglia correlates with seizure duration and frequency [35]. However, it is still unclear to what extent neuroinflammation contributes to the development of epileptic seizures, even though it is clearly not a mere epiphenomenon [36].

In 2016, in resected brains from FCD patients, it was shown that microglial activation could be partially caused by CD47/SIRP-α- and CD200/CD200R-mediated reductions in the immune inhibitory pathways, where chronic neuroinflammation has been observed [37].

Recently, in eight children affected by FCD type II, the inflammatory molecule signaling of high-mobility group box 1 (HMGB1) through Toll-like receptor 4 (HMGB1-TLR4), was found to be altered. Pro-inflammatory cytokines downstream to HMGB1 were upregulated in tissues coming from the resected area compared to those from the perilesional zone [38], even if this did not correlate with the severity of epilepsy. 

On the contrary, some studies demonstrated a positive correlation between two biomarkers of neuroinflammation, respectively, human leukocyte antigen-DR isotype (HLA-DR) and IL-17, and the frequency of seizures per month (Figure 2; [35,39]).

However, it is not known if there is a correlation between remission from epilepsy and reduction in neuroinflammation, and such a study would be particularly important to predict remission in childhood epilepsy [40].

However, mechanistic studies are limited by the lack of animal models for FCD reproducing all the features of the human disease, and by the poor access to human samples. In the last few years, new animal models based on genetic manipulations that better simulate the human pathology have been produced [41]. Some of these models rely on the use of in utero electroporation to alter the proteins of the mTOR pathway [42,43,44]. Therefore, it is likely that in the future, more insights can be gained on the functional connection between upregulated inflammatory molecules and epileptic seizure genesis/progression, in order to find better biomarkers for the prognosis of this disease.

### 2.2. Genetic Epilepsy: PCDH19 Epilepsy

A possible involvement of neuroinflammation has been proposed for PCDH19 epilepsy. This syndrome is characterized by seizures starting in early infancy, as well as behavioral problems, intellectual disability, and developmental delay [45,46]. It is caused by mutations in the X chromosomal gene Pcdh19 and, interestingly, affects heterozygous females, while hemizygous males are spared [45]. The seizures show sensitivity to fever and tend to occur in clusters. The sensitivity to fever has led researchers to hypothesize that the immune system might be involved in seizure generation [47]. Therefore, Higurashi et al. set out to investigate the efficacy of corticosteroid application to treat seizures in five PCDH19 epilepsy patients. In all five patients, seizures ceased rapidly, even if the effects were transient. Two case reports followed Higurashi’s work, confirming the rapid cessation of seizure clusters in PCDH19 patients upon corticosteroid administration [48,49]. Interestingly, Higurashi et al. reported one patient to whom corticosteroids were successfully administered prophylactically. Also, the patient from Lee and Chung’s study showed no recurrence of seizures for at least three years after corticosteroid treatment [49]. These results warrant research into the prophylactic use of corticosteroids in PCDH19 epilepsy.

The pathogenesis of PCDH19 epilepsy is still unclear. To explain the efficacy of corticosteroids in treating the disease, Higurashi et al. speculated about BBB disruptions as a possible pathogenic mechanism. BBB disruptions are known to interfere with brain homeostasis and trigger seizures. Corticosteroids can improve BBB integrity and thereby protect against seizures [50]. Indeed, such a mechanism of action could explain why PCDH19 epilepsy patients respond so rapidly to corticosteroid treatment. Further support for this hypothesis comes from the detection of antibodies against several epitopes of the NMDA receptor in the cerebrospinal fluid (CSF) of patients after seizures. Such antibodies indicate a non-specific immune response against neuronal proteins that were degraded during the seizure. Their presence in the CSF suggests that they leaked from the brain into the CSF, through a weakened BBB. Finally, PCDH19 is highly expressed in endothelial cells of the brain [51], and it is possible that in heterozygous females PCDH19 mosaic expression interferes with BBB integrity. Future research in mouse models of PCDH19 epilepsy should elucidate whether BBB disruptions are a feature of the syndrome and what molecular pathways are involved in this pathogenesis. 

### 2.3. Brain Tumors: Glioblastoma Multiforme

Glioblastoma multiforme (GBM) is the most common and lethal primary brain tumor, with an average life expectancy of 12–15 months. The standard treatment procedures, combining surgery, radiation, and chemotherapy, are not capable of contrasting it [52]. Due to its robust vascularization and invasive properties, its five-year survival rate is only about 3.3% [53]. GBM has been classified into four different subtypes, recognized as neural, pro-neural, mesenchymal and classical GBMs [54,55]. Inflammation spreads in all subtypes of GBM, but it is more profound in the mesenchymal subtype, which, interestingly, has the worst prognosis [56].

GBM is associated with hyper-excitability and seizures at different rates [57,58]. Together with other brain tumors, GBM is the second most common cause of focal intractable epilepsy [59,60].

The following three types of mechanisms have been involved in explaining high GBM invasiveness and proliferation: (1) alteration of the microglia [61], (2) changes in the GBM microenvironment throughout the activation of cytokines cascades, and (3) BBB disruption and angiogenesis [62].

Firstly, microglia play an important role in immune surveillance and represent the largest tumor-infiltrating cell population, contributing up to 30% of the total tumor mass. The gene expression pattern of microglia interacting with GBM cells is altered. GBM can circumvent the neuroimmune system by lowering its tumor-sensing abilities, suppressing immune responses, and promoting tumor invasion processes [63]. It has been seen that upon hyperactivation, mTOR signaling promotes cell proliferation and metabolism, and thereby contributes to this tumor initiation and progression [64]. 

Next, the tumor microenvironment sustains tumor progression by releasing inflammatory cues in response to its dynamic interaction with endogenous cells [65,66,67,68]. Waters et al. identified two cytokines, IL-1β and oncostatin M (OSM), that activate Re1B/p50 complexes through the NF-κB pathway, causing an increase in pro-inflammatory cytokines in GBM cells and a worse prognosis in patients [69]. Likewise, a study observed a significant rise in GBM cell proliferation promoted by IL-1β-induced ERK activation [70], while another one reported increased levels of IL-1β receptors in human GBM cell lines (U87MG) overexpressing epidermal growth factor receptor variant III (EGFRvIII) [71]. It is known that downstream of IL-1β, IL-6 and IL-8 promote tumor growth and invasion [72,73,74]; specifically, IL-6 seems to be a poor prognostic factor for patients’ survival [75]. Indeed, IL-6 sustains angiogenesis, acting on endothelial cells and astrocytes through the activation of vascular endothelial growth factor (VEGF) and fibroblast growth factor (FGF) [71]. Another role for IL-6 released by microglia was recently shown in vitro, where IL-6 contributes not only to angiogenesis, but also to increased interstitial fluid pressure, edema formation, and changes in blood flow, ultimately causing defective drug delivery [76].

Indeed, pro-inflammatory cytokines are thought to trigger BBB disruption [77] and this is one of the main causes of brain tumor-related edema (BTRE). BTRE is one of the main causes of the mortality of GBM, since the accumulation of fluids within the rigid skull rapidly increases the intracranial pressure, which can result in decreased cerebral flow, ischemia, brain herniation, and death [78]. Both in animal models and human patients affected by GBM, a beneficial effect of treatment with corticosteroids (e.g., dexamethasone) has been shown. Indeed, this treatment reduces peritumoral fluid, even though the exact mechanism for its action is poorly understood. The following two explanations have been suggested: a direct action of dexamethasone on the vessel components, or a more general anti-inflammatory action that might counter the effect of inflammatory cytokines in BBB breakdown [79].

So, neuroinflammation represents the principal cause of high GBM proliferation and could also impinge, at the same time, on the altered brain excitability. Further studies need to be done in this direction. 

### 2.4. Epilepsy Caused by Infection: Maternal Immune Activation

Converging lines of evidence from basic science and clinical studies suggest a relationship between perturbations **of the maternal** immune system during pregnancy and neurodevelopmental disorders such ASD and schizophrenia [80]. Indeed, maternal immune activation (MIA) can trigger seizures in the offspring [81]. MIA can be induced in animal models by the administration of either LPS **or polyinosinic**:polycytidylic acid (PIC) to pregnant females. It is worth noting that MIA can cause astrogliosis [82]. In a recent paper [83], in a model of LPS-induced MIA, the administration of the pro-epileptic drug pentylenetetrazol (PTZ) to the offspring caused seizures in these mice, once adult, which were more severe and frequent, and an enhancement of anxiety-like behavior compared to the control mice. Then, these MIA mice were more vulnerable to the cognitive impairment caused by treatment with PTZ. Finally, since the inflammatory cytokines TNF-α and IL-10 are involved in MIA, the authors measured the presence of these proteins in the hippocampus of the offspring, and they found that MIA significantly enhanced TNF-α and IL-10 production there.

Previously, in a different model of MIA (obtained by injection of PIC), Washington and colleagues [84] found that after prenatal exposure to IL-6, seizures induced by kainic acid (KA) administration were less frequent. Instead, co-exposure to IL-6 and IL-1β in the same time window increased seizure frequency. Also, other studies have shown a synergic role of IL-6 and IL-1β in regulating hyper-excitability [85].

Another molecular explanation on the mechanisms involved in MIA-induced epilepsy was recently provided by Corradini [86]. Indeed, a single PIC injection at embryonal stage nine was sufficient to increase offspring susceptibility to seizures at three months of age (P90). In contrast, PIC administration in adult mice did not increase the susceptibility to seizures. Even if neuroinflammatory markers, as the number of microglial (Iba+) cells and the activation of CD11b and GFAP, were found to be normal, the authors found an alteration in network activity. In this model, prenatal exposure to pro-inflammatory molecules produced a delay in the normal excitatory-to-inhibitory switch of the neurotransmitter GABA. Indeed, during physiological cortical development, the effect of GABA release changes from excitatory to inhibitory [87]. In this model, prenatal exposure to neuroinflammation changed ionic transporter NKCC1 and KCC2 expression, and consequently intracellular chloride concentration; treatment with the NKCC1 antagonist bumetanide rescued the phenotype. GABA was shown to retain an excitatory effect in acute slices of adult mice in this MIA-induced epilepsy model. This suggests a delay in the excitatory/inhibitory switch of chloride, as already shown for numerous other pathological models, including spinal cord lesions, chronic pain, brain trauma, cerebrovascular infarcts, autism, Rett and Down syndrome, various types of epilepsies, and other genetic or environmental insults [88].

### 2.5. Autoimmune Neurodegenerative Disease: Multiple Sclerosis

Multiple sclerosis (MS) is a neurodegenerative disease of the CNS, characterized by the demyelination of neural axons, mediated by the immune system [89]. MS is the most frequent cause of non-traumatic neurological disability in young adults. Its onset is generally between 20 and 40 years; MS signs and symptoms can vary in severity, and include motor impairment, tremor, vertigo, weakness, pain and vision loss [90].

Although its cause is still unknown, this autoimmune disease has a multifactorial origin, and it seems to be associated with several genetic and environmental factors. Some factors are variants of genes involved in the immune response, such as HLA, IL2RA and IL7RA, which are also related to an increased risk of developing other autoimmune diseases [91].

MS is manifested with typical lesions of the nervous tissue, called focal plaques, characterized by the de-myelinization of axons. The extension and inflammatory activity in these lesions can vary depending on the form of MS. Usually active lesions present a high level of neuroinflammation, and there is a local disruption of BBB and a high density of immune cells as lymphocytes, activated microglia and macrophages. Indeed, analogously to GBM, also in MS, the protective effect of BBB is disrupted, and this allows pro-inflammatory factors and immune cells circulating in the blood stream to reach the CNS and contribute to inflammation [92]. Moreover, a physiological alteration of astrocytes with pronounced astrogliosis has been found in models of MS, with perturbed expression of astrocytic molecules controlling brain homeostasis and excitability, as water channels (AQP4) and synaptic glutamate transporters (EAAT2) [93].

Notably, an increase in the prevalence of epileptic seizures has been observed in MS patients [94,95]. The increase is about three to six times compared to the general population and the link between the two pathologies is not well understood [96,97].

In particular, in a Swedish population, a meta-analysis study showed a correlation between the severity of MS (including increased duration, increased disability, and progressive, as opposed to relapsing remitting disease) and the appearance of epilepsy [98]. Moreover, lesions in the brain and spinal cord of MS patients can cause involuntary movements that look like seizures. These seizure mimics are called non-epileptic seizure-like activity, and can be recognized by a lack of signature in EEG, and they should be treated differently from seizures [99]. Published studies on the presence of epileptic seizures in MS have been focused mostly on the epidemiology and treatment [94,96,100,101]. Importantly, some treatments, such as interferons, baclofen, and aminopyridines acting to limit MS symptoms, can elicit epilepsy and should be applied in a case-by-case modality. 

Thus, considering the role of neuroinflammation in both MS and epilepsy, it can be worthwhile to further investigate the role of pro-inflammatory factors in seizures affecting MS patients. This could be a starting point for the development of new treatments or better management of already known therapeutics.

### 2.6. A New Infection Associated to Altered Brain Excitability with Unknown Etiology: SARS-CoV-2

Emerging evidence suggests that the coronavirus SARS-CoV-2, the etiologic agent of coronavirus disease 2019 (COVID-19), can cause neurological complications, even though its main target is the respiratory system [102]. An increasing amount of research suggests an association of SARS-Cov-2 to hypoxic/ischemic encephalopathy, acute cerebrovascular disease, and impaired consciousness in hospitalized Chinese patients in Wuhan [103,104]. Moreover, neurological widespread symptoms, including encephalopathy, prominent agitation and confusion, and corticospinal tract signs, have been found in a cohort of French patients [105]. Finally, epilepsy has been correlated with SARS-CoV-2- induced neuroinflammation [106,107,108,109].

SARS-CoV-2 is a coronavirus (severe acute respiratory syndrome-coronavirus-2) that emerged in late 2019 in China, and it is characterized by severe respiratory problems that are often associated with gastrointestinal infections. Often starting like a normal flu, it can fastly degenerate attacking lungs. Until now, it is the third among the CoVs to have been originated in these last 20 years (together with CoVs causing SARS in 2002/2003 and MERS in 2012), but it is surely the most aggressive. Indeed, SARS-CoV-2 is responsible for an ongoing pandemic.

Although studies testing whether SARS-CoV-2 targets the brain in humans or animal models are not yet available, it is well established that other CoVs, such as CoV-OC43, [110,111,112,113,114] target the CNS and cause neurological alterations, including brain inflammation and encephalomyelitis. If SARS-CoV-2 was indeed to target the brain, there could be important long-term consequences on the CNS. Research has demonstrated that chemokine presence in the brain, caused by prolonged inflammation, can contribute to chronic illness, neurodegeneration, psychiatric disease, and epilepsy [115,116]. 

Anosmia and taste loss, associated with COVID-19, also raise some of the following interesting questions: are they caused by a peripheral effect of SARS-CoV-2 (e.g., on the olfactory nerve, taste receptors), or are they related to a direct effect on the CNS? Additionally, evidence was found that SARS-CoV-2 could enter the CNS by crossing the neural–mucosal interface in the olfactory mucosa [117]. Therefore, the mechanisms of action and entry of SARS-CoV-2 in the brain should be extensively investigated, to better understand its correlation with its temporary or long-lasting neurological effects. It is known that SARS-Cov-2 binds to cells via the S1 subunit of its spike protein. Rhea and co-workers [118] have now demonstrated that intravenously injected radio-iodinated S1 (I-S1) readily crossed the BBB in male mice, entering the parenchymal brain space. I-S1 was also taken up by the lung, spleen, kidney and liver. Intranasal administration of I-S1 also entered the brain, although at levels roughly ten times lower than intravenous administration.

The investigation on how COVID-19 affects the brain is still ongoing, but neurological alterations appear to be common in COVID-19 patients; a meta-analysis reported that 96.1% of COVID-19 patients present an abnormal EEG [119]. The percentage of COVID-19 patients that developed de novo seizures was reported to be low by some studies (0.47–0.66% [104,120]; some cases have been reported of patients showing high levels of inflammation and epileptiform brain activity [121]. Notably, some patients with epilepsy experienced an increase in seizure frequency [122]. This could be caused by other environmental factors, such as the enhanced level of emotional distress and anxiety that these patients had to face during the COVID-19 pandemic. Indeed, many patients with epilepsy perceived a lack of care towards their condition [122]. Interestingly, some patients with epilepsy have reported a decrease in the frequency of their seizures, and this could be caused by the isolation and reduction in events that trigger seizures [123].

Apart from the indirect effects of the CODIV-19 pandemic on patients with epilepsy, a direct effect of COVID-19 on the increase in seizures and on de novo seizures cannot be excluded. The possible mechanism is still to be uncovered, but some possible factors have been found. In COVID-19 patients, an increase in the production of pro-inflammatory cytokines (TNF-α, IL-6, IL-1B) had been reported [124], which is linked to neurotoxicity and epilepsy. COVID-19 infection also causes a break-down of the BBB [124]. As already stated, such a disruption can cause osmotic imbalance in the brain, and allows immune cells and peripheral pro-inflammatory cytokines to enter the central nervous system. Indeed, the BBB disruption, as we have already seen in previous paragraphs, appears to be common in neural pathologies that show co-morbidity with epilepsy, such as PCDH19 epilepsy, glioblastoma and multiple sclerosis.

Neurologists should be aware that autoimmune neurologic complications involving the CNS might occur, and neurological complications should be promptly recognized and treated to reduce permanent neurologic disability, as in the case of acute disseminated encephalopathy [125] or of Guillain-Barré syndrome, which is an acute dysimmune neuropathy [126]. 

Finally, although most people that were affected showed no or mild symptoms upon SARS-CoV-2 infection, the severity increases in elderly people [127]. In aged subjects, neuroinflammation can be potentiated by the activation of pre-existing or new inflammatory cascades activated by COVID-19, as well as by a stressful lifestyle caused by the pandemic, creating a vicious circle that leads to a global increase in mortality [127].

### 2.7. Alterations in Gut Microbiota in Epilepsy and in Autism Spectrum Disorders with Co-Morbidity with Epilepsy

In these last years, a relationship between the brain and the gut microbiota has been proposed, which can ultimately be involved in brain inflammation through the still poorly understood relationship between immune responses and the CNS. Postnatal brain development is paralleled by the maturation of the ecosystem of symbionts populating the gastrointestinal (GI) tract. Interestingly, it has been seen that possessing a healthy microbiome might play a key role in correct neurodevelopment during the early stages of life [128]. On the contrary, in many neurological diseases, such as in ASD and epilepsy, an alteration in gut microbiota has been shown [129,130,131,132,133,134].

Moreover, several alterations in gut microbiota associated with intestinal problems have been found in other neuropsychiatric disorders of potential neurodevelopmental origins, such as schizophrenia [128,135,136,137], bipolar disorder [138] and depression [139,140], where the balance between excitation and inhibition is impaired [141].

Recently, it has been shown that in Shank3 knock-out (KO) mice, a model of Phelan McDermid syndrome (PMS), a form of autism associated with drug-resistant epilepsy, there is a significantly different GI morphology associated with alterations in the microbiota composition in feces, which may contribute to inflammatory responses [142]. This happens because Shank and other synaptic proteins are also expressed in enterocytes [143]. Moreover, a change in inflammatory cytokine levels was reported [144], and higher E. coli LPS expression in liver samples of PMS, together with an increase in IL-6 and activated astrocytes [142]. Dysbiosis of the microbiota in ASD is consistent with a disruption of the intestinal permeability, with consequent influences on the interaction between the gut and brain [145]. A paper by Möhle and co-workers [146] has shown that the immune response is a gateway between commensal bacteria in the GI trait and the CNS. Indeed, after antibiotic treatment, reduced neurogenesis is produced by the decreased infiltration of specific immune cells, i.e., Ly6Chi monocytes. Conversely, if the Ly6Chi population is treated with probiotics and exercise, or if Ly6Chi monocytes are adoptively transferred in animals treated with antibiotics, there is a rescue of hippocampal neurogenesis. Then, in a model of kindling, where stress facilitates epileptogenesis, it has been demonstrated that this effect is mediated by the microbiome [147]. Another study shows a change in the microbiome in patients with focal idiopathic epilepsy with respect to healthy subjects [148]. Finally, it has been shown that a ketogenic diet can be helpful in treating epilepsy, because it can have an impact on microbiome regulation [149].

## 3. An Overview on Therapies Affecting Neuroinflammation with Possible Outcome in Epilepsy

Finally, on the basis of the interaction between epilepsy and neuroinflammation that we have illustrated in general, and for some exemplificative pathologies, in this section we summarize several therapeutic strategies directed to contrast neuroinflammation and impinging on amelioration of the epileptic phenotype (Figure 3). Many of these drugs have been selected on the basis of results from preclinical studies in different animal models of epilepsy [150], and thanks to the discovery of several biomarkers in patients [151]. This section, far from being exhaustive on this issue, tries to recapitulate principal results in the treatment of epilepsy, also evidencing some problems related to the use of drugs that stop neuroinflammation. 

### 3.1. Inhibition of Immune Response

First of all, neuroinflammation involves complex interactions between innate and adaptive immunity, as seen in animal models [153]. The main goal of immunotherapy is to reduce acute inflammation and minimize irreversible neuronal dysfunction. People with autoimmune epilepsy may usually have clinical problems in addition to their seizures, including psychiatric difficulties, cognitive problems, balance impairment, sleep disorder, and autonomic (involuntary actions such as breathing or heartbeat) dysfunction, which should be treated apart. Interfering with these molecular cascades has been seen as beneficial in treating epilepsy with several of the following different drugs:– Corticosteroids and especially adrenocorticotropic hormone (ACTH) have been used in various forms of childhood epilepsy, as in the case of PCDH19 female epilepsy ([47]; see the previous paragraph);– Antagonists of Toll-like receptor have been efficaciously used in convulsive epilepsy, as in the case of resveratrol, an anti-inflammatory stilbenoid. The application of resveratrol reduced the frequency of spontaneous seizures in KA-treated rats [154]. This effect was associated with a reduction in neuronal cell loss and an inhibition of mossy fiber sprouting in the hippocampus;– Inhibition of the prostaglandin E2-PGE2- receptor subtype is neuroprotective in a pilocarpine model of SE [155];– Immunosuppressants, such as cyclosporine A, FK-506 (also known as tacrolimus), and rapamycin inhibiting T-lymphocyte activation, can stop seizures [156]. Indeed, daily systemic injection of cyclosporine A or FK-506 during electrical amygdala kindling prevented the acquisition of severe chronic seizures in rats [157]. However, long-term protection from crisis failed after drug withdrawal, showing limited anticonvulsant capacities [158]. Moreover, these data are quite controversial, since other authors showed opposite effects [159];– Immunoglobulins (IVIg) have been first employed in intractable epilepsy, starting from the empirical observation of its beneficial effect on seizures [160]. Indeed, immune system dysfunction could trigger, maintain or, unexpectedly, sustain intractable seizures [161];– In status epilepticus, minocycline represents a promising candidate for the anti-inflammatory treatment of epilepsy [162,163]. Despite often being referred to as an inhibitor of microglial activation, minocycline also affects—either directly or indirectly—other cell types, such as neurons, astrocytes, and oligodendrocytes. A similar drug, Minozac, blocks the production of pro-inflammatory cytokines and prevents the cognitive degenerative phenotype associated in a mouse ‘two-hit’ model of epilepsy [164]. Interestingly, an IL-1β inhibitor, VX-765, being used in psoriasis therapy, completed phase 2 clinical trials in 60 people with treatment-resistant partial-onset epilepsy [165].

### 3.2. Antibody Antagonists

Monoclonal antibodies against immune cell membrane proteins, such as efalizumab and natalizumab, already used in autoimmune pathologies such as psoriasis, multiple sclerosis and Crohn’s disease [166], have been used to target serum auto-antibodies in epileptogenesis [167]. Some ameliorations have been found after early treatment with immunomodulatory therapies, in autoimmune encephalitis and autoimmune epilepsies. Recently, autoantibodies against the IL-1 blockade are proposed for refractory epilepsy in an adolescent female with pharmaco-resistant epilepsy with a good outcome [168]. Then, tocilizumab, a humanized monoclonal antibody against the IL-6 receptor, has been found to be successful in a case report of two patients with pediatric refractory status epilepticus and acute epilepsy [169]. It is clear that there is the lack of standardized researches in this field, despite evidences emerging from basic research. The reason is that it is often difficult to find the right target of patients to be treated, due to the fact that the critical step is to intervene as early as possible [161].

### 3.3. Probiotics

According to the results in animal models, the use of probiotics, prebiotics (as probiotic nutrients) and dietary manipulations, such as a ketogenic diet, could be promising to regulate homeostasis in brain excitability [170]. However, standardized studies with controlled administration of probiotics need to be done to better investigate this issue [171,172]. For example, a recent clinical trial reported a beneficial effect of probiotics in a pilot study, with a reduction in seizures to a 50% level in about 30% of the subjects, and a general amelioration of life quality [173].

### 3.4. Cannabinoids

The endocannabinoid system (ECS) has been shown to contribute to neuroinflammation, and neuroinflammation can cause epilepsy; consequently, there is much evidence about the positive use of cannabinoids to treat chronic epilepsy [7]. Indeed, ECS can modulate the balance between excitation and inhibition, through the release of endogenous cannabinoids (endocannabinoids). Specifically, during neuroinflammation cannabidiols inhibit the activity of cyclooxygenase-2 (CO-X 2), 5-lipoxygenase and cytochrome P450, reducing the expression of inflammatory molecules, such as prostaglandins and leukotrienes [174]. Interestingly, a drug belonging to this class, Epidiolex, has been very recently approved by the FDA as a treatment for pharmaco-resistant epilepsy in Dravet syndrome [175]. Moreover, the use of phytocannabinoids, i.e., active molecules present in *Cannabis sativa*, has been effective in a wide range of pathologies with neurological correlates, among which are chronic pain, nausea, and multiple sclerosis. However, more studies are needed in this field, since cannabidiols and its derivatives are still considered illegal in several countries, and because a secondary collateral effect of phytocannabinoids cannot be neglected in many cases [174]. 

### 3.5. Inhibitors of Voltage-Gated Potassium Channels Kv1.3

Another therapeutical strategy to block neuroinflammation is the use of inhibitors of voltage-gated potassium channels Kv1.3 [176]. Importantly, abnormal expression of Kv1.3 channels has been demonstrated to also be correlated with epilepsy [176]. Toxins produced by sea anemones, scorpions, spiders, snakes, and cone snails can target specific subsets of T lymphocytes as well as microglial cells, acting on their Kv1.3 channels and blocking neuroinflammation. Clinical trials to explore the efficacy of this treatment are currently ongoing, even if a problem could be the difficulty for these venoms to pass BBB.

## 4. Conclusions

Patients with drug-resistant epilepsy and many animal models of epilepsy show active inflammation [177]. So far, pediatric patients with refractory seizures that are resistant to common anti-epileptic drugs have been mostly treated with drugs counteracting neuroinflammation [178,179]. What appears evident is that drugs acting on various inflammatory pathways can mitigate the epileptic phenotype, but the response is both subject-based and dependent on the type of pathology causing epilepsy, as we have seen from this overview. Blocking neuroinflammation can be especially effective in counteracting the cascade mechanisms of recurrent seizures. As a future perspective, it would be important to explore if a pretreatment with anti-inflammatory drugs could block the emergence of seizures in subjects that are prone to epilepsy because of genetic diseases, brain trauma, tumors, infections, or SARS-COV2.

Further studies in this field are necessary in order to understand if neuroinflammation is a signature or a cause of epilepsy, or both, in order to better orient the time-course of therapies and to standardize protocols involving anti-inflammatory treatments.

## Figures and Tables

**Figure 1 ijms-22-06981-f001:**
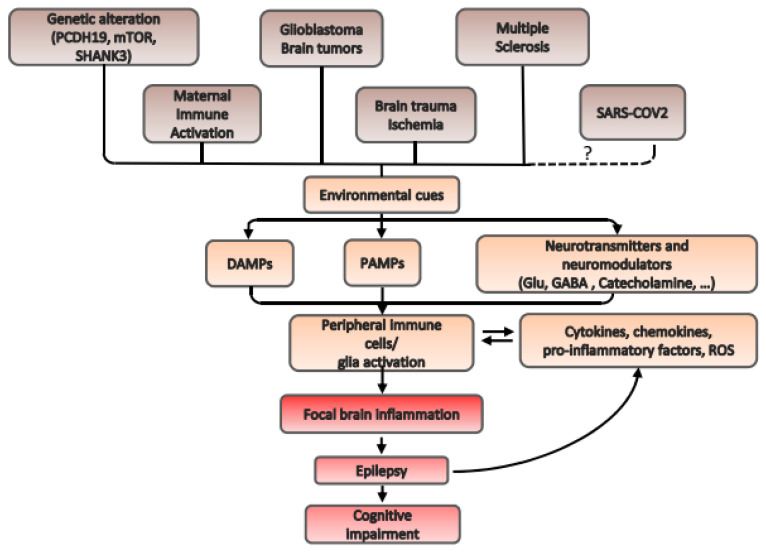
Schematic diagram of the interaction between epilepsy and neuroinflammation. Epilepsy is related to neuroinflammation and neuroinflammation can induce epilepsy in a biunivocal interaction. Many molecular mechanisms have been described to be involved in this loop. Diverse mechanisms, here summarized in the diagram, can impinge by means of the interaction with environmental cues on damage-associated molecular patterns (DAMPs), pathogen-associated molecular patterns (PAMPs) or on neurotransmitters and neuromodulators. Dotted lines and question mark indicate the possible involvement of SARS-COV2 in this loop; studies about its mechanism of action are currently ongoing.

**Figure 2 ijms-22-06981-f002:**
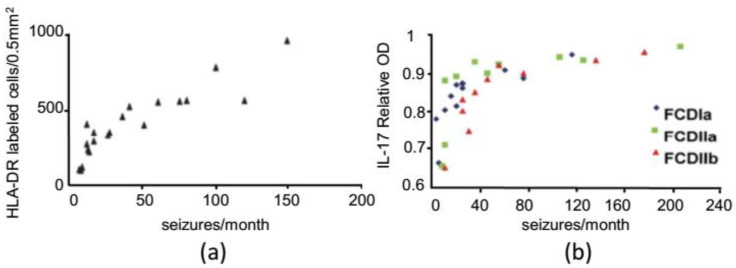
Correlation between inflammatory pathways and frequency of seizures in two exemplificative studies done in FCD patients. (**a**) Positive correlation between the distribution of cells of the activated microglia/macrophage lineage (HLA-DR positive cells) and frequency of seizures (replotted with permission from [35]). (**b**) Positive correlation between IL-17 positive cells and frequency of seizures in three different types of FCD (IL-17 relative optical density is expressed in y-axis; replotted with permission from [39]).

**Figure 3 ijms-22-06981-f003:**
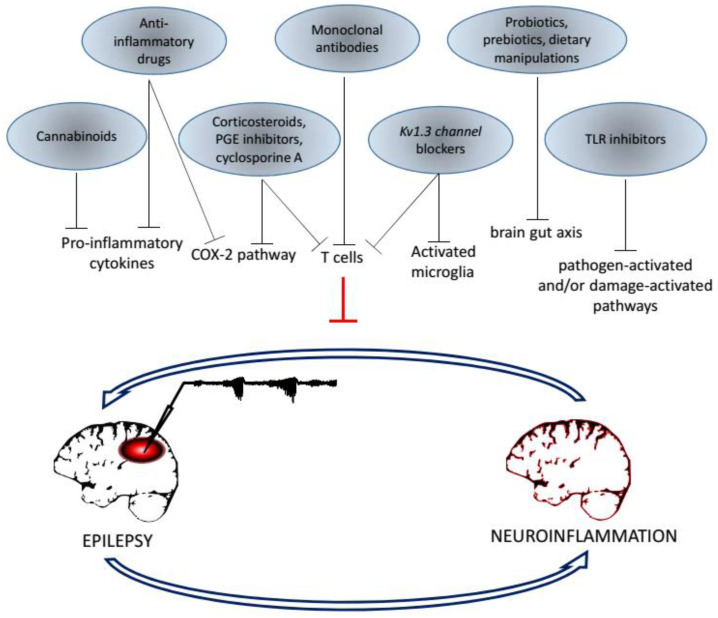
Schematic representation of attempted therapeutic strategies targeting neuroinflammation and capable to ameliorate epilepsy. The diagram illustrates the interaction with specific families of drugs and their targets; their main function seems to interrupt the loop that reinforces neuroinflammation produced by epileptogenesis. The trace recorded from the left brain is a typical example of critical activity due to the localized treatment with the convulsive agent 4-AP in our experiments, as conducted in [152].

## Data Availability

We followed MDPI Research Data Policies.

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
