# Peer review of "Neuroinflammation: A Signature or a Cause of Epilepsy?"

_ijms, 2021, doi:10.3390/ijms22136981_

Round 1

Reviewer 1 Report

The Authors, Drs Pracucci et al., submitted a review in which they attempt to explore and discuss the relationship between epilepsy and inflammation in some representative neurological and systemic disorders.

The manuscript is scientifically interesting. Paragraphs are well organized, and the purpose of the article is clear.

However, there are some comments that need proper answer:

1) The section “Mechanisms of neuroinflammation in some exemplificative pathologies” should include also multiple sclerosis, as the more relevant neuroimmunological disease.

2) An additional attractive color figure should be included;

3) The bibliography should be updated. Several references are old;

4) The English language needs revisions.

Author Response

Response to reviewer#1: We are grateful to reviewer#1 for their comments about our review. In particular:

  • We agree with this point and we added a section about the connection between multiple sclerosis and epilepsy in the section “Mechanisms of neuroinflammation in some exemplificative pathologies related to epilepsy”.
  • We propose a possible new figure as Figure 3 to summarize the part relative to possible therapeutical interventions used to contrast neuroinflammation and that ameliorate epilepsy. We have inserted this figure at the end of this section. We think it is the only attractive figure we can present, considering we show already a diagram relative to the cellular/physiological pathway that link neuroinflammation and epilepsy. We hope it could respond to reviewer’s request. To insert this new figure, we moved Figure 2 at the beginning of the section “Mechanisms of neuroinflammation in some exemplificative pathologies related to epilepsy”, introducing its presentation. We have introduced opportunely in the text also Figure 3 with a new sentence.
  • We eliminated/updated some references and where there were old references, we put in tracking changes modality ‘new citation’ within the manuscript.
  • Since moderate English changes were required, we asked our native speaking English authors Vinoshene Pillai and Didi Lamers to revise carefully the manuscript.

Reviewer 2 Report

This is interesting review article that should be published after a small revision that should focus on one point:

- the Authors should add information that among drugs that can potentially be used in treatment of epilepsy are inhibitors of voltage-gated potassium channels Kv1.3. These compounds might be used in treatment of neuroinflammatory diseases, among them also epilepsy, via inhibition of Kv1.3 channels expressed in activated T lymphocytes and microglial cells, for details see review by Wang et al., Frontiers in Neuroscience, 2020, vol. 13, Article 1393, doi: 10.3389/fnins.2019.01393.

Author Response

We thank reviewer#2 for their comments. We added a paragraph within the ‘Therapeutic perspectives’ section about the use of Kv1.3 channels blockers to block neuroinflammation and consequently epilepsy referring in particular to clinical ongoing trials about this topic. About revisions for English style, we did moderate English revisions, as they were requested by referee#1, in particular about fine/minor spell check.

Round 2

Reviewer 1 Report

No additional comments

Author Response

no additional comments to reply; we did minor spell check.